# Physiological Basis for Using Vitamin D to Improve Health

**DOI:** 10.3390/biomedicines11061542

**Published:** 2023-05-26

**Authors:** Sunil J. Wimalawansa

**Affiliations:** Medicine, Endocrinology & Nutrition, Cardio Metabolic Institute, (Former) Rutgers University, North Brunswick, NJ 08901, USA; suniljw@hotmail.com

**Keywords:** 25(OH)D, 1,25(OH)_2_D, autocrine and paracrine, endocrine, human, mechanisms, musculoskeletal, morbidity and mortality, public health

## Abstract

Vitamin D is essential for life—its sufficiency improves metabolism, hormonal release, immune functions, and maintaining health. In contrast, vitamin D deficiency increases the vulnerability and severity of type 2 diabetes, metabolic syndrome, cancer, obesity, and infections. The active enzyme that generates vitamin D [calcitriol: 1,25(OH)_2_D], CYP27B1 (1α-hydroxylase), and calcitriol receptors (VDRs) are distributed ubiquitously in cells. Once calcitriol binds with VDRs, the complexes are translocated to the nucleus and interact with responsive elements, up- or down-regulating the expression of over 1200 genes and modulating many metabolic and physiological functions. The administration of vitamin D_3_ or its correct metabolites at proper doses and frequency for extended periods will achieve the intended benefits. The same principle applies to designing and conducting RCTs. Different tissues have varied thresholds for 25(OH)D concentrations. To mitigate conditions such as infections/sepsis and cancer and reduce premature deaths, levels above 50 ng/mL are necessary. Cholecalciferol (D_3_) (not its metabolites) should be used to correct vitamin D deficiency and raise serum 25(OH)D to the target concentration. Calcifediol [25(OH)D], in contrast, increases serum 25(OH)D concentrations rapidly and boosts the immune system in one day. Thus, it is the agent of choice in emergencies such as infections, for those in ICUs, and insufficient hepatic 25-hydroxylase (CYP2R1) activity. Calcitriol is crucial to maintaining serum-ionized calcium concentration in persons with advanced renal failure and hypoparathyroidism. Since administered calcitriol does not enter immune cells, it is ineffective in other conditions, like infections; it should not be used as vitamin D replacement therapy. Considering the high costs and higher incidence of adverse effects due to narrow therapeutic margins (ED50), 1α-vitamin D analogs, such as 1α-(OH)D and 1,25(OH)_2_D, should not be used for other conditions. Similarly, calcifediol costs 20 times more than D_3_: it should not be used as a routine vitamin D supplement for hypovitaminosis D, osteoporosis, or renal failure. Healthcare workers should resist accepting inappropriate promotions, such as calcifediol for chronic renal failure and calcitriol for osteoporosis or infections—there is no physiological rationale for doing so. Standard diets have too little vitamin D. Maintaining the population’s vitamin D sufficiency (above 40 ng/mL) (and individuals above 50 ng/mL) with vitamin D_3_ supplements and/or daily sun exposure is the most cost-effective way to reduce chronic diseases and sepsis, overcome viral epidemics and pandemics like COVID-19, and reduce healthcare costs. Furthermore, vitamin D sufficiency improves overall health, reduces absenteeism, improves productivity, reduces the severity of chronic diseases such as metabolic and cardiovascular diseases and cancer, decreases all-cause mortality, and minimizes infection-related complications such as sepsis and COVID-19-related hospitalizations and deaths. Properly using vitamin D is the most cost-effective way to reduce chronic illnesses, infections, and healthcare costs; thus, it should be a part of routine clinical care.

## 1. Introduction

From a physiological point of view, humans are expected to generate more than 80% of their vitamin D requirements through ultraviolet sunrays, as their diet intake is minimal. Significant behavioral changes have occurred over the past few decades; more people are working indoors and avoiding the sun. Consequently, clinically relevant vitamin D deficiency has become a global pandemic. Vitamin D deficiency increases the global burden of acute and chronic diseases and healthcare costs.

In those with vitamin D deficiency, increasing 25(OH)D concentrations via D_3_ supplements or sufficient daily sun exposure reduces the risks and the severity of multiple disorders, including type 2 diabetes, hypertension, metabolic syndrome, obesity, cancer, and infections [1,2,3]. Despite this, there is little consensus from randomized controlled clinical trials (RCTs) regarding the proper intake and optimum serum 25(OH)D levels for preventing the mentioned complications. This lack of agreement is primarily due to the poorly designed studies; many clinical studies piggybacked on evaluating pharmaceuticals, failure to include vitamin D insufficient or deficient subjects in RCTs, infrequent doses administered, participants are allowed to take over-the-counter supplements (including vitamin D) during clinical trials, and the failure to measure circulatory 25(OH)D concentrations to ensure the target level is achieved.

Adverse effects from vitamin D supplements—hypercalcemic syndrome due to over-dosing of vitamin D—are extremely rare. When they occur, they are invariably due to taking extremely high (i.e., supra-pharmacological) doses mistakenly, too frequently [4,5]. Vitamin D toxicity should not be diagnosed in isolation—incidental findings of elevated serum 25(OH)D concentrations should correlate with clinical signs and symptomatology. Characteristics of the calcitriol-driven hypercalcemic syndrome include serum 25(OH)D concentration over 150 ng/L (over 375 nmol/L) associated with hypercalcemia (raised serum ionized calcium), hypercalciuria (urinary calcium, exceeding 400 mg/24 h), and suppressed parathyroid hormone (PTH) concentration [6]. All three components must be present to diagnose vitamin D toxicity—an exceedingly rare event in the community.

### 1.1. Vitamin D_3_ (Cholecalciferol)

Vitamin D is an essential micronutrient that is essential for human survival. Humans are expected to generate most of their vitamin D requirement from exposure to ultraviolet B rays (UVB) from sunlight. D_3_ is synthesized in the skin after exposure to UVB rays, while D_2_ originates from plants. UVB photons from UVB rays photolyze 7-dehydrocholesterol in the skin to produce previtamin D_3_, which subsequently isomerizes to form vitamin D_3_. The cycle of the generation of vitamin D and its activation steps leading to synthesizing 25(OH)D and 1,25(OH)_2_D is illustrated in Figure 1.

Over-exposure to UVB rays can cause erythema and skin damage and must be avoided. However, excess sun exposure does not overproduce vitamin D or cause hypercalcemia. Due to an inherent evolutionary feedback mechanism, the synthesis of previtamin D is inhibited. Excess D_3_ generated is photodegraded in the skin, thus preventing it from entering circulation and reaching the liver. While excessive unprotected sun exposure could harm the skin, it will not lead to pathologically high vitamin D in circulation [3,7]. Free-living hunter-gatherers and lifeguards in swimming pools and beaches are exposed daily to plenty of sunlight. They maintain an average serum 25(OH)D concentration of 46 ng/mL (a range of between 40 and 65 ng/mL) [3,8,9].

The physiological range of 25(OH)D is between 40 and 80 ng/mL, encompassing normal circumstances and under free-living conditions. Acute illnesses, chronic stress conditions (e.g., post-traumatic stress disorders), chronic inflammatory diseases like fibromyalgia), and infections significantly consume vitamin D, magnesium, B2, and other cofactors. Therefore, for optimal recovery, these need to be replaced promptly during these conditions. When there are extenuating circumstances and stresses, such as infections (endemics/pandemic), comorbidities, metabolic or cardiovascular diseases, or cancer, higher serum 25(OH)D levels are necessary for the precursors of calcitriol [D and 25(OH)D] to enter peripheral target cells like immune cells to generate calcitriol within these cells to overcome the situation (see Section 1.4 and Section 4.1, for details).

With the activation of cytochrome P450, the CYP2R1 gene increases the expression of 25-hydroxylase enzyme in the liver, producing 25(OH)D. The latter is further hydroxylated by the 1α-hydroxylase enzyme, derived by the CYP27B1 gene in the mitochondria in the endoplasmic reticulum, to form 1,25(OH)_2_D (calcitriol) in renal tubules as well as in peripheral target cells, such as immune cells. Calcitriol production in renal cells is regulated by PTH and a negative-feedback control by FGF-23 (secreted by osteoblasts) and 1,25(OH)_2_D (Figure 1). However, this negative-feedback control is minimal in peripheral target cells.

### 1.2. Fundamental Benefits of Vitamin D

Irrespective of the reason for hypovitaminosis D or the condition, vitamin D_3_ is the choice for replacement therapies to maintain serum 25(OH)D concentration at the desired levels and to maintain a robust immune system [10]. When exposed to high densities of bacteria or viral loads, as with SARS-CoV-2, adhering to straightforward public health measures is crucial to prevent the spread of the infection. No drugs or nutrients are available except for traditional vaccines to prevent infections. However, such preventative vaccines are not available against SARS-CoV-2.

In contrast, maintaining a robust immune system with sufficient vitamin D, although it may not prevent infections, substantially reduces the development of symptomatic disease and complications from infections. Furthermore, it significantly lowers community transmissions of infections and outbreaks. In addition to preventing symptomatic disease, a robust immune system would notably reduce hospitalizations, ICU admissions, complications, deaths, and healthcare costs.

In 2011, raising the RDA for all age groups by the US Institute of Medicine (IoM) was a step in the right direction but was grossly inadequate [11]. Moreover, suggestions by the IoM [11] are public health recommendations for authorities and are not guidance for individuals or diseases. Besides, those suggestions do not apply to persons outside North America [3,8,9,12,13,14]. However, because of the ambiguity of such narrowly focused governmental reports, such publications were misinterpreted by organizations and misled governments and the public outside the USA. Because of uncertainties, they assumed reports like IoM as recommendations for the public. Consequently, the implementation of publicized misleadingly minute doses and lower serum 25(OH)D concentrations than required for better human health have harmed those populations.

Currently, most nutrient RDAs in many countries are outdated. They are insufficient to raise serum 25(OH)D even beyond 20 ng/mL—a level considered inadequate). Such is too little to overcome acute or chronic non-skeletal diseases and infections or promote good health [15,16]. Ironically, the currently recommended vitamin D food fortification guidelines are also outdated [17,18] and should be increased by three- to four-fold [19].

### 1.3. Vitamin D Metabolites and Analogs and Their Clinical Uses

There are specific indications for using vitamin D metabolites and analogs—calcifediol or 1α-hydroxylated compounds. Synthetic analogs of vitamin D are twenty-fold more expensive than vitamin D_3_. Since the ED50 of these metabolites and analogs is narrow, unsurprisingly, they have a higher incidence of adverse effects. Consequently, vitamin D analogs are inappropriate for use as long-term vitamin D supplements, as in cases of vitamin D deficiency and osteoporosis [20]. Calcifediol (25-hydroxylated vitamin D) is only indicated in those with defective 25-hydroxylation, as in hepato-cellular failure, severe malabsorption syndromes, and when the serum 25(OH)D concentrations need to be raised rapidly in emergencies, such as infection [20].

Intestinal vitamin D absorption occurs via the lymphatic system as chylomicrons bind to VDBP and then 25-hydroxylated in the liver. This takes two to five days in a healthy person and over a week in an ill person [21]. In contrast, calcifediol is already 25-hydroxylated and absorbed directly into the venous system without delay. Therefore, it enters the systemic circulation within four hours of ingestion [21,22]. Considering this, calcifediol is a reasonable choice for emergencies and for those with malabsorption syndromes.

Inefficient 1α-hydroxylation is present in advanced renal failure and rare genetic disorders, such as pseudo-hypoparathyroidism and hypoparathyroidism. Such individuals require 1α-hydroxylated vitamin D metabolites to maintain serum ionized calcium concentration in the lower physiological range. These valuable lifesaving agents improve the quality of life in advanced renal failure, hypoparathyroidism, and pseudo-hypoparathyroidism [20].

Magnesium sufficiency is crucial for many biological activities, including hormone synthesis and release (e.g., insulin and PTH) and calcitriol with vitamin D (calcitriol) receptor (VDR) interactions [23,24]. The proper biological functions of CYP450 enzymes, vitamin D, and vitamin D (calcitriol) receptors (VDRs/CTRs) depend on the availability of the correct intracellular concentrations of magnesium [23,25] and other co-factors. Furthermore, magnesium adequacy reduces the complications and mortality from post-COVID syndrome [26] and chronic kidney diseases [27,28,29]. Notably, magnesium and 25(OH)D are consumed due to metabolic activities during infections. Therefore, higher maintenance doses of these cofactors are necessary during such stress periods.

### 1.4. Consequences of Hypovitaminosis D

Because of the negative genomic control process, severe vitamin D deficiency is associated with hyperinflammation, oxidative stress, and autoimmunity [30,31]—an over-reactive pathological immune response [32]. This could lead to cytokine storms [33,34], causing acute respiratory distress syndrome (ARDS). When circulating D_3_ and/or 25(OH)D is adequate (e.g., over 50 ng/mL), these precursors will diffuse into peripheral target cells, such as immune cells, in adequate quantities from the circulation, allowing the generation of sufficient calcitriol intracellularly, which suppress multiple pathological processes. In addition to genomic effects, these crucial biological processes occur via autocrine and paracrine signaling mechanisms [35,36]. This reduces the risks of cytokine storms and ARDS and is associated with severe pulmonary and cardiovascular complications in persons with COVID-19 [37,38].

Hypovitaminosis impairs intracrine and paracrine signaling, thus weakening the immune system and increasing vulnerability to infections and complications [36,39]. The pro-inflammatory and oxidative stress responses associated with cytokine storms in severe viral infections increase the need for intensive care unit (ICU) admissions and the risk of death. Children with serum 25(OH)D concentrations less than 12 ng/mL (i.e., severe vitamin D deficiency), when infected with SARS-CoV-2, could develop life-threatening hyper-inflammatory conditions such as Kawasaki-like disease or multi-system inflammatory syndrome [40,41,42].

Moreover, the weakened adaptive immunity in hypovitaminosis D reduces the generation of neutralizing antibodies, impairs the cytotoxic action of immune/killer cells, reduces the effectiveness of memory cells and macrophages, and causes weaker responses after (any) vaccination. In those with a more fragile immune system (primarily due to severe hypovitaminosis D), SARS-CoV-2, infection or immunization could lead to significant adverse effects, including autoimmune reactions, generalized hyper-inflammation, and pathological oxidative stresses, which could lead to systemic complications and death. Consequently, in 2020, due to the prevailing high incidence of hypovitaminosis among older people, COVID-19 became a pandemic, primarily affecting those with severe vitamin D deficiency [43,44].

In addition, calcitriol-mediated effects in immune cells facilitate the development of anti-microbial peptides, neutralizing antibodies, stabilizing epithelial and endothelial cells, and strengthening gap junctions [45]. These broader membrane-stabilization effects minimize fluid leakage (e.g., preventing pulmonary edema) and viral spread into soft tissues [3,46,47,48]. To mitigate the widespread vitamin D deficiency, a personal vitamin D response index (which is not validated) has been suggested to optimize vitamin D supplementation (or safe sun exposure). However, the associated cost and lack of practicality dampen its use. A better and more cost-effective option is adhering to broader population recommendations [42], which leads to population vitamin D sufficiency with minimal cost [44,49].

### 1.5. Intracellular Synthesis of Calcitriol and Benefit from Vitamin D

CYP27B1 enzyme is present in all endocrine cells, such as pancreatic islets and parathyroid, thyroid, testes, ovary, and placental cells [50], in which calcitriol is synthesized intracellularly; it also modulates hormone release and metabolic functions [51]. In addition, vitamin D adequacy enhances DNA repair, suppresses malignant cell development [52,53,54], and improves the prognosis of malignancies [55,56]. Proper vitamin D intake is inversely associated with the incidence of lung cancer [56]. Sufficient circulatory D and 25(OH)D concentrations (i.e., above 50 ng/mL) [10] lead to better physiological responses, such as defense against microbes [57], the prevention of autoimmune diseases such as multiple sclerosis and inflammatory bowel disease [58,59], and the synthesis and secretion of hormones [51], controlling the renin-angiotensin-aldosterone system (RAS) and the FGF23/klotho system [3,7].

Since vitamin D is a nutrient, adequate supplements significantly benefit those with deficiency [60]. However, giving extra vitamin D to those already sufficient (e.g., serum 25(OH)D concentration above 40 ng/mL) will unlikely provide additional benefits (with some exceptions discussed below). These exceptions include those with cancer [61,62] and infection [63,64], such as septicemia and SARS-CoV-2 [64,65]—these require the maintenance of serum 25(OH)D concentrations higher than 40 ng/mL.

In addition to the ability to generate 1,25(OH)_2_D, the presence of a high density of VDRs in epithelial cells allows them to optimize barrier functions and other biological activities. Examples include the lungs [66], breast [67], intestine [68], prostate [69], osteoblasts, and chondrocytes [70,71]. Unsurprisingly, all these tissues are targets for calcitriol. The inactivation of VDRs in chondrocytes reduced the expression of FGF23 in osteoblasts. It increased the expression of 1α-hydroxylase and sodium phosphate cotransporter type Il in proximal renal cells—another mechanism to increase calcitriol production [70]. These facilitate the modulation of chondrocyte-mediated signaling (e.g., FGF23) in renal tubular cells and for bone cell formation and osteoid calcification.

### 1.6. Why This Study Is Important and Necessary

Due to the lack of agreement between different medical and scientific societies and the negative contribution from the IoM, there is apathy and confusion about the necessary oral doses and the optimum serum 25(OH)D concentrations. In addition, many healthcare workers are unfamiliar with the proper circumstances and dose for using vitamin D metabolites and their benefits vs. risks. Despite the crucial biological functions of vitamin D, textbooks in immunology do not even discuss the role of vitamin D (or nutrients) in the immune system. Missing such fundamental education impedes advancing knowledge and practice to benefit patients, even among specialists such as immunologists.

These ambiguities have led to recommendations for using sub-optimal doses by scientific societies, erroneous advice provided to government bureaucrats by nutrition-related committees like IoM and the USPTO in the USA, the scientific advisory committee (SCAN) and National Institute for Health and Care Excellence (NICE) in the UK, dampen the creation of updated vitamin D guidelines [48,72,73]. In addition, the use of high doses of vitamin D intermittently—infrequent intervals of more than two weeks intervals instead of using daily or once a week D_3_, inappropriately prescribing expensive synthetic vitamin D analogs, and grossly exaggerating rare adverse effects of vitamin D leads to fear-mongering of using this safe nutrient are additional examples. These errors in clinical practice recommendations and myths prevent the betterment of the population. This study aims to overcome some of the mistakes and myths mentioned above.

The recommending of sub-optimal (standard-outdated) vitamin D doses to everyone, ignoring their body weight (and obesity and other variables affecting their serum 25(OH)D concentrations), or not using serum 25(OH)D-based calculation of the proper doses for individual persons [10,20], has led to using pediatric doses of vitamin D in adults with no benefit to recipients. This manuscript also illustrates examples and circumstances where the efficacy, clinical requirement to raise serum 25(OH)D rapidity (as in emergencies such as sepsis, SARS-CoV-2-infection, at ICU set-up, etc.), and the target serum 25(OH)D concentration necessary to overcome the underlying illness such as infection or cancer needs (see Section 3.4).

The content of the manuscript is primarily based on published evidence. Over four decades of practical and clinical experience and research on vitamin D by the author and over fifty peer-reviewed publications related to vitamin D enrich his perspective. Data was gathered from literature searches on scientific databases, including PubMed and the Library of Congress (Medicine). The all searches were personally conducted by the author, accumulating and synthesizing data, generating figures, and creating a list of pertinent references. The author has no conflicts of interest—he did not receive pharmaceutical, writing, or external assistance or input in developing this manuscript.

## 2. Vitamin D Generation—Genomic and Non-Genomic Actions

The primary hormonal function of calcitriol is maintaining calcium homeostasis and the conservation of calcium and minerals. Humans have developed multiple evolutionary mechanisms using calcitriol that enhance intestinal calcium absorption, calcium conservation in renal tubules, and dynamic interaction on skeletal calcification and resorption (Figure 1). In conjunction with parathyroid hormone (PTH), calcitriol profoundly affects renal tubular calcium reabsorption [74]. Calcitriol has a broader regulatory function and maintains tight calcium-phosphate homeostasis through VDR-genomic interactions and non-genomic membrane effects. Collectively, these actions prevent fluctuations of serum-ionized calcium, keeping a narrow range. These effects are summarized in Figure 1.

### 2.1. Genomic Actions of Calcitriol

Calcitriol is an essential regulator of calcium-phosphate homeostasis and participates in tightly regulated VDR genomic activity. In conjunction with PTH, the hormonal calcitriol promotes osteoid calcification and maintains skeletal integrity, thus effectively preventing rickets and osteomalacia [75]. The endocrine action of calcitriol facilitates musculoskeletal functions, balance, and coordination and improves reflexes via nerve conduction, resulting in an overall improvement in muscular functions [76]. Genomic actions of calcitriol are slow to initiate but are associated with biological outcomes over a longer duration.

When the concentration of 1,25(OH)_2_D increases above the physiologic threshold, it increases the expression of CYP24A1, 24-hydroxylase enzyme, and the production of FGF-23. This feedback system also inhibits PTH production, thus reducing the generation of 1,25(OH)_2_D. It simultaneously increases the expression of CYP24A1, increasing the catabolism of 25(OH)D and 1,25(OH)_2_D into their respective inactive 24-hydroxylase metabolites: 24,25(OH)D and 1,24,25(OH)_2_D [50]. The 24-hydroxylation-mediated catabolic products of calcifediol and calcitriol (Figure 2) have no known biological actions. In contrast, low circulatory calcitriol suppresses CYP24A1 and stimulates the CYP27B1 gene to produce the hormonal form of calcitriol from the kidney. Calcitriol and its inactive catabolic derivatives are illustrated in Figure 2.

As with some other micronutrients, vitamin D represents another example with potential use as a personalized or targeted approach to disease prevention, minimizing complications from critical illnesses, including infection. However, the target circulator levels of individuals vary (depending on the target tissue, disease, and underlying vitamin D status) for achieving therapeutic goals [3,7] (see Section 3.6 for details). Vitamin D-related metabolomics, transcriptomics, and epigenetics studies likely provide paths for better critical outcomes [77,78].

The most biologically active form of vitamin D, 1,25(OH)_2_D, regulates over 1200 genes within the human genome, and gene polymorphisms and epigenetics would further influence its mechanism of action [30,79,80]. Meanwhile, the up-regulation of gene transcription leads to the release (stimulation) of anti-inflammatory and antioxidant cytokines, and the down-regulation (suppression) of inflammatory cytokines leads to a cascade of beneficial effects [35,57,81,82,83].

### 2.2. Non-Genomic Actions of Calcitriol

In addition to VDRs in the cytosolic compartment, VDRs are also located in cell membranes [84]. The internalization of vitamin D and its D-binding protein (VDBP) complex is another example of rapid membrane responses [85]. This internalization occurs in specific cells, such as renal tubular, fat, and muscle cells, enabling active transportation. This transpires via the megalin with its co-receptor, cubilin [86]. In addition, vitamin D influences mitochondrial functions directly and indirectly, including fusion, energy production, modulation of mitochondrial membrane potential, regulation of ion channels, and apoptosis [85].

In contrast to genomic actions, non-genomic responses to vitamin D are rapid but last for a shorter duration [85]. Calcitriol triggers rapid calcium influx through epithelial cells (membranes) and uptake by various cells, such as gastrointestinal, renal tubular, osteoclast, and osteoclast cells. Some of these actions occur via its membrane vitamin D receptor and the protein disulfide-isomerase A3 (PDIA3), responsible for some rapid non-genomic responses [85]. Calcitriol also stimulates the release of second massagers via membrane receptors, which modulate several intracellular processes. The latter include the control of cell cycle, proliferation, and immune responses through wingless (WNT) [87], sonic hedgehog (SSH), STAT1-3, or NF-kappaB pathways [88]. These calcitriol-mediated rapid actions facilitate calcium spikes that modulate signaling and multiple other physiological pathways [89].

Despite advances, the exact mechanisms of non-genomic responses to vitamin D are not fully understood. Vitamin D also, directly and indirectly, influences mitochondrial function, including fusion-fission, energy production, mitochondrial membrane potential, the activity of ion channels, and apoptosis. However, the mechanisms of the non-genomic responses to vitamin D are still not fully understood [85].

## 3. Clinical and Randomized Control Studies (RCTs)

Well-designed and statistically powered RCTs that used proper doses of vitamin D as the primary interventional agent have all reported favorable clinical outcomes. These positive clinical outcomes are not restricted to the musculoskeletal system—rickets and osteomalacia [90], but also apply to other body systems [79,80], disorders, and diseases [3,91,92,93]. The main issue with the clinical trials performed to date is that a few have been explicitly designed to test the hypotheses that hypovitaminosis is a causative factor and have specific vitamin D-related endpoints. In addition, the designs of many clinical studies lack the targeting required to achieve pre-specified 25(OH)D concentrations. In contrast, all well-designed studies with sufficient subjects (statistical power) in persons with hypovitaminosis D using the proper doses for an adequate duration have reported favorable clinical outcomes [94,95,96].

### 3.1. Contributions from Recent Clinical Studies

Many studies have confirmed the association between vitamin D intake and the reduction of viral respiratory infections [97,98], including in persons with COVID-19 [99,100,101,102]. Further, more than 120 peer-reviewed clinical studies used vitamin D as the primary intervention to investigate COVID-19. Clinical outcomes from these studies [103] were published on the following URLs [https://c19early.org/d] [103]. This site provides all negative and positive published studies on vitamin D and COVID-19 [103]. This data has been made available with unrestricted access, making this dataset more valuable than all RCTs, meta-analyses, and systematic reviews, especially considering some meta-analyses are poorly designed, biased, and address a few cherry-picked RCTs.

Although it is not essential to fulfill all criteria, vitamin D fulfills all of Hill’s causation criteria [104]: hypovitaminosis D significantly increases vulnerability, causing complications and deaths from COVID-19. Vitamin D- and COVID-19-related data published in over 390,000 subjects should have been examined in 2020 using a Big Data meta-analysis. That would have provided additional validation with a larger effect size and higher statistical power to make firm conclusions.

### 3.2. Reasons for Failure of Recent Vitamin D Randomized Controlled Clinical Studies

During the last few years, a few negative clinical studies were published related to vitamin D. The failures of these highly publicized prominent RCTs have been amplified by dozens of duplicate publications by the same and other authors using the same database; the VITAL study is one such example [105]. Failed clinical outcomes are directly related to poor study designs. These include unfamiliarity with the biology and physiology of vitamin D and/or how vitamin D benefits the body system (e.g., mechanisms of vitamin D on disease prevention and stimulating the immune system). In some cases, study investigators’ lack of understanding of biology or conflicts of interest has led to designing studies with failed outcomes.

It is important to note that there is no rationale in expecting studies to demonstrate the beneficial effects of vitamin D supplementation/treatment in people who are not deficient. Despite these, studies such as VITAL and others recruited subjects who were not vitamin D deficient. Whether such flawed study designs are used mistakenly, due to ignorance of the biology and physiology of vitamin D, or deliberately downplay the importance of this unpatented, natural nutrient is unclear. Table 1 characterizes components of a well-designed nutrient/vitamin D clinical study/RCT.

Disregarding vitamin D as a nutrient and treating it as a pharmaceutical agent is another major fallacy of several recent RCTs. These fundamental errors, unfortunately, exist in recent larger, sponsored clinical studies, including the VITAL study [105,107]. RCTs designed correctly and administered age- and condition-appropriate vitamin D doses consistently improve the intended clinical outcomes. Examples include insulin sensitivity [42], facilitating weight loss [109], reducing cardiovascular risks [53,110], improving renal functions [111], and skeletal disorders [112].

### 3.3. Other Critical Aspects to Consider, Specifically for Vitamin D-Related RCTs

If a loading dose is not used at the entry into a clinical study to raise serum 25(OH)D rapidly, it could take weeks or months to achieve 50 ng/mL concentration. Studies related to preventing chronic health issues, clinical trials must be a few years long. Examples are cancers, chronic kidney disease, Alzheimer’s and Parkinson’s, chronic kidney disease, cardiovascular disorders, macular degeneration, cataracts, increased bone mineral density, fractures, and osteoporosis.

In addition, several health problems arise from the harmful effects of maternal hypovitaminosis D on fetuses. Therefore, RCTs designed to provide vitamin D as an intervention in infants and children are unlikely to be entirely successful—i.e., too late to intervene effectively. As mentioned above, virtually none of the acute studies have taken account of the higher utilization (rapid consumption) of vitamin D and magnesium in acute illnesses like infections (e.g., COVID-19). Thus, outcomes may not be optimal. In such studies, the interventional group taking standard doses is unlikely to benefit significantly—they need higher doses of vitamin D and cofactors.

In addition, in cancer, diabetes, obesity, multiple sclerosis, etc., vitamin D receptor activity is down-regulated, requiring higher amounts of available (free) 25(OH)D. This is one of the main reasons these diseases need more elevated serum 25(OH)D concentrations than usually accepted, as illustrated in Figure 3. There is also emerging evidence that providing loading doses could overcome VDR down-regulation. In conditions like malabsorption syndromes, gastrointestinal absorption of oral D_3_ is significantly delayed or markedly impaired. These situations can be overcome by replacing D_3_ with calcifediol [10,20].

### 3.4. Examples of Larger Vitamin D Interventional RCTs with Significant Study Design Errors

In many recent clinical studies, including the VITAL study (2000 IU/day; ∆ ~10 ng for 5.3 years) [113,114], vitamin D assessment (ViDA) study (∆ ~28 ng: 200,000 IU, monthly doses) [115], D2d study (cancer and pre-diabetes) [116], vitamin D to improve outcomes by leveraging early treatment (VIOLET) RCT (∆ ~35 ng: a single dose of 40,000 IU in critically ill patients) [117], and vitamin D on all-cause mortality in heart failure (EVITA) study (∆ ~16 ng: 4000 IU/day) [118], all had significant study design errors. In several of these, as with VITAL [114,119], the control groups were allowed to take over-the-counter supplements [107], including vitamin D [105,107,108]. This reduces the effect size between the active and placebo groups (or the control group), resulting in a loss of statistical power and erroneous non-conclusive outcomes (type 1 error).

In addition, the clinical outcomes from vitamin D clinical studies, including RCTs conducted for a too-short duration and the administration of too-low and/or too-infrequent doses, are unreliable, and conclusions are misleading. Consequently, irrespective of the study size, the cost, credibility, or where it was conducted (e.g., university name), clinical outcomes from poorly designed studies (see Table 1) are unreliable and should not be generalized or used for policy decision-making. RCTs that fail to adhere strictly to the nutrient-related essential criteria and research principles mentioned in Table 1 will fail to generate useful or meaningful data. Detailed guidelines for conducting nutrient clinical trials and systematic reviews have been reported [48,120,121].

### 3.5. Contrasts between Negative and Positive RCTs

The main reason for RCTs’ outcome failures in critically ill subjects, as in ICUs, is that raising circulating concentrations of 25(OH)D in seriously ill patients would take more than a week—another example of a lack of insight into basic biology while designing RCTs leading to poor outcomes. In seriously ill patients, even very high doses of vitamin D, such as those over 400,000 IU, administered as a large single [117,122] or intermittent doses will have no beneficial effects. Therefore, favorable (expected) outcomes cannot be anticipated in RCTs with major design flaws. In contrast, a systematic review analysis of six vitamin D intervention trials in acute SARS-CoV-2 infection reported a significant relative risk reduction (RRR) of 0.60 (95% CI 0.40 to 0.92, *p* = 0.02), and the rates of RT-CR positivity reduced significantly in the intervention group, with a RRR of 0.46 (95% CI 0.24 to 0.89, *p* = 0.02) [123].

As described in this article and others recently [10,20], these significant study design errors were due to the lack of understanding of the biology of vitamin D and the mechanisms of calcitriol modulating body systems. In the case of subjects in the ICU, failed outcomes are predictable with any dose of vitamin D, even in those with severe vitamin D deficiency. Eliminating such outcome failures is straightforward: administering the correct type of vitamin D—calcifediol instead of vitamin D (see Section 4.6 for rationale and dose recommendation) [99,100,102]. Hence, it is important to have a deeper understanding of the biology of different vitamin D metabolites.

RCTs with study design errors fail to demonstrate benefits (i.e., no significant differences of primary endpoints from placebo) in diabetes, cancer, osteoporosis, cardiovascular endpoints, mortality, etc. However, subgroup outcome analyses of most of these studies using serum 25(OH)D concentrations as a denominator demonstrated beneficial positive endpoints. This reflects the lack of insights and weaknesses of these RCTs. In parallel, some meta-analyses reported beneficial clinical outcomes when multiple failed RCTs, despite their failure individually. This suggests that these individual RCTs did not have the statistical power to differentiate outcomes in active vs. control groups. Such examples include reductions in fractures and/or falls, early recovery from respiratory tract infections, reduced severity of cancer, and low death and all-cause mortality rates [107].

### 3.6. What to Expect from RCTs

Before accepting the results and conclusion of any nutrient RCT, one must carefully evaluate (A) the study design, (B) protocol, (C) adherence to the stipulated (pre-defined) procedures, (D) statistical methods used, and (E) conflicts of interest and sponsorships. Several recent and ongoing large RCTs with significant study design errors are not exceptions to generate meaningful or clinically useful information. These studies are unlikely to generate meaningful or valuable data. RCTs not designed to test a primary hypothesis, those lacking robust clinical study protocols and/or statistical power to make meaningful conclusions, would not fill the gaps in the knowledge.

Apart from five RCTs related to vitamin D and COVID-19 with poor study designs, over 120 other clinical studies, including 41 RCTs [124,125], provided a larger effect size and strong associations between vitamin D and clinical outcomes of COVID-19 [103]. These studies provided overwhelming evidence that vitamin D deficiency fulfills Bradford Hill’s criteria for causality—“hypovitaminosis D significantly increases vulnerability, causing complications and deaths from COVID-19” [10,20,126,127]. Other studies with robust protocols reported statistically significant clinical outcomes, such as with multiple sclerosis [10,20,126,127], periodontal disease [10,20,126,127], and cancer [Munoz, 2022], especially against important cancer types [128].

### 3.7. Systems and Different Tissues May Need Different Serum Concentrations of 25(OH)D

It is noteworthy that the circulatory 25(OH)D concentrations required to overcome disease conditions, such as drug-resistant migraine/cluster headaches, psoriasis, asthma, etc., and during infectious pandemics and endemics, they are higher than the generally recommended concentrations [10,20]. Published data over the past two decades suggest the possibility of tissues/body systems related to different circulatory levels of 25(OH)D for optimal function [3]. A summary of conclusions from many clinical studies is illustrated in Figure 3.

Higher circulatory 25(OH)D and D_3_ concentrations are crucial for maintaining a robust immune system to impede viruses such as SARS-CoV-2 [64]. Nevertheless, prevention efforts should be directed at the “entire person”, not at a given tissue or a system. Furthermore, conditions and diseases (e.g., infections) can manifest quickly, precluding the opportunity to optimize serum 25(OH)D concentrations. Therefore, the prevailing data strongly suggest that maintaining serum 25(OH)D concentrations above 50 ng/mL (125 nmol/L) as the best option for the broadest protection [10,20].

## 4. Pharmacodynamics and the Underlying Mechanisms of Calcitriol

Calcitriol acts via multiple mechanisms, benefiting humans. It activates the innate and adaptive immune systems and has broader genomic and non-genomic actions. Several key biological and physiological functions of calcitriol, including autocrine and paracrine signaling mechanisms, are mediated from intracellularly synthesized 1,25(OH)_2_D (calcitriol) in peripheral target cells [129], and not the circulating hormonal form of calcitriol, of which the concentrations are far too little to enter peripheral target cells. This locally generated calcitriol in these peripheral target cells (e.g., immune cells), such as macrophages, T- and B-lymphocytes, and dendritic cells, provide essential signaling for biological functions, such as autocrine and paracrine signaling mechanisms of vitamin D and prevention of autoimmunity [129] (see next Section).

### 4.1. Autocrine and Paracrine Signaling

In addition to the direct effects of calcitriol on the genome, it also exerts indirect effects via autocrine and paracrine signaling [20]. Examples of these include the impact of calcitriol switching T helper cell 1 (Th1) to T helper cell 2 (Th2) and Th17 to Treg cells, which transforms pro-inflammatory status to anti-inflammatory status [35,36]. When the intracellular calcitriol concentration is low, statutes of Th1 and Th17 cells remain inflammatory, contributing to cytokine storms and the development of ARDS following viral infections in vulnerable people [130,131]. In addition, intracellular calcitriol in immune cells, directly and indirectly, enhances the expression of anti-microbial peptides and antibody synthesis [132,133].

While the exact mechanism of stimulation of some of the above pathways is unclear, it is known to involve transcription factors C/EBPβ and inhibit the orphan receptor NR4A2 [134]. The regulation of the CYP27B1 gene (1α-hydroxylase enzyme) by a transcriptional factor promoter, NR4A2, is inhibited by C/EBP-beta. Furthermore, over-expression of C/EBP-beta decreases NR4A2 and CYP27B1 mRNA levels [134]. In contrast, FGF-23 counteracts the 1α-hydroxylase enzyme through FGF receptors in the presence of the co-receptor (an aging-related factor), Klotho [3]. At the same time, the ablation of Klotho leads to the over-expression of the FGF23 phenotype, which is consistent with Klotho deficiency [3]. This signaling also activates the mitogen-activated protein kinase (MAPK) cascade, but its role in CYP27B1 expression remains unclear [135].

Despite the above, there is little evidence from RCTs regarding the optimum serum 25(OH)D levels for preventing various disease-related complications. This confusion derives from the non-standardized clinical studies using different serum 25(OH)D concentrations, correlated with minimum effective concentrations, and poorly designed and conducted RCTs [107] (see Section 3). The fundamental flaw is that investigators failed to regard vitamin D as a nutrient; thus, when designing RCTs, vitamin D should not be considered as a synthetic pharmaceutical agent. Clinical outcome failures are common, as the pharmacokinetics of these two sets of agents are very different (Figure 3). Furthermore, they failed to adhere to published guidelines on nutrient clinical studies (Table 1) [120].

### 4.2. Optimum Concentrations of Circulating 25(OH)D Concentrations

As illustrated above (Figure 3), tissue-specific serum concentrations of 25(OH)D are used to elicit biological effects. The high affinity of vitamin D and 25(OH)D to the VDBP provides the means for transporting and enhancing the half-life of 25(OH)D in circulation. In addition, the specific evolutionary mechanism of the megalin–cubilin-mediated membrane-driven internalization of vitamin D and 25(OH)D molecular complexes provides the energy-dependent entry of these compounds into renal cells and vitamin D and other nutrient storage tissues [136,137]. This entry facilitates the synthesis of the hormonal form of calcitriol in renal tubular cells and storage mechanisms for D_3_ and 25(OH)D in muscle and fat cells. In contrast to pharmaceuticals, these mechanisms allow 25(OH)D to have a naturally longer circulatory half-life [138,139]. Administration of a nutrient such as vitamin D in a deficient person leading to a clinically meaningful change in nutrient status (∆), as measured with serum 25(OH)D concentration (see Section 3.3), would produce a measurable, statistically significant clinical outcome. However, that is present only in those with nutrient-deficient status.

In contrast to micronutrients like vitamins, iron, iodine, etc., pharmaceutical agents neither have a specific internalization process nor a physiologic storage mechanism—a significant difference, and their actions are short-lived. Consequently, pharmaceutical agents need to be administered once or usually multiple times daily to maintain the necessary serum concentrations to achieve the intended benefits [140,141]. The formula of pharmaceutical agents has been modified (e.g., changed to precursors or slow-release forms, converted into longer-acting formulae like depot forms) so as the frequency of administration to overcome their too-shorter actions to prolong their half-lives and activities. Consequently, the dose-response curves for the two mentioned categories differ vastly. Therefore, one cannot evaluate these two categories using the same principles, protocols, or study designs. Figure 4 illustrates the fundamental differences between (4A) circulatory concentrations of a nutrient and (4B) a pharmaceutical agent.

As illustrated above, many recent larger vitamin D RCTs have been poorly designed for bizarre reasons. Some were designed to fail, and others piggybacked on industry-sponsored studies as a shortcut to generating data without additional expenses. Their primary endpoints focus on pharmaceutical agents, not nutrients. In such studies, vitamin D-related outcomes are incidental or secondary; thus, they cannot be relied upon for drug approvals, guidelines, or policy decision-making. In addition, the doses, frequency of administration, and duration of studies were inappropriate in many vitamin D RCTs. As a result, their endpoints and conclusions were unreliable.

### 4.3. Importance of Raising Population 25(OH)D Concentrations to Reduce Morbidities

The guidelines of the US Endocrine Society [the minimum adequate serum 25(OH)D of 30 ng/mL (75 nmol/L)] were for individual patients, not for disease entities or special or vulnerable groups [16]. In the absence of adequate exposure to sunlight, to raise and maintain a blood level of 25(OH)D above 30 ng/mL, individuals require a daily minimal oral intake of 2000 to 4000 IU (50 to 100 µg) of vitamin D_3_, with a longer-term safe upper limit of 10,000 IU [124,142,143,144,145]. Different scientific organizations have recommended varied serum 25(OH)D concentrations. However, the minimum level to be maintained to overcome infections, cancer, autoimmunity, and heart disease and for robust immunity is 50 ng/mL [10,20,57,63,146].

The natural vertebrate-form of vitamin D_3_ (the parental vitamin D) is the desirable supplementation, preferably obtained via routine safe skin exposure to ultraviolet sun rays and/or daily or weekly supplementation or targeted food fortification programs [14]. These mentioned modes can provide sufficient amounts of vitamin D to maintain the population’s vitamin D sufficiency [i.e., maintaining the mean serum 25(OH)D concentrations above 40 ng/mL) [21,49], which would enhance the population’s immunity and significantly reduce illnesses and absenteeism, thus increasing productivity. A robust immunity in the population can inherently curtail the spread of pathogenic microbial infections, particularly viral epidemics and pandemics such as SARS-CoV-2, and reduce the associated hospitalization and deaths from infections and diseases [64,147].

### 4.4. Factors That Modify the Functions of CYPP450 Enzymes and VDRs

The quantity and the efficiency of generating vitamin D in the skin and the subsequent two hydroxylation steps are reduced by several factors. These include the location of residence (i.e., latitude) where no UVB rays reach during winters (i.e., seasonality), air pollution that blocks UVB rays, sun-avoidance behavior such as excessive clothing, using umbrellas and UV blockers, higher melanin skin pigmentation, scarring or aging of the skin, etc.

While VDR abnormalities such as pleomorphism are not uncommon [148], genetic variants of CYP450 vitamin D-related enzyme abnormality are rare. For the proper functioning of CYPP450 enzymes, it is crucial to have adequate intracellular concentrations of not only calcitriol but also magnesium [149,150]. In addition, diseases, especially infections, increase vitamin D consumption, thus requiring a higher intake [151,152]. Furthermore, acute and chronic inflammation increases reactive oxygen species (i.e., generating oxidative stress) that reduce mitochondrial functions [153], which further aggravates oxidative stress and cytochrome 25- and 1α-hydroxylase activity and impair recovery [154].

The continual availability of sufficient D_3_ in conjunction with the pleiotropic distribution of VDRs, co-factors, and the associated signaling systems enables a healthy life [49,155]. The vitamin D control system—CYP450 enzymes that activate and degrade vitamin D and metabolites, PTH and FGF23—associated feedback mechanisms, and VDRs have evolved over millions of years [156]. This complex system has been fine-tuned through vertebrate evolution, resulting in genetic variants and nucleotide polymorphisms in modern humans [157,158]. Calcitriol–VDR interactions and the resultant gene expressions related to target metabolic genes and subsequent biological activities are further modulated by epigenetic variations [30,158] and switching from traditional lifestyles to modern [159].

### 4.5. The Importance of Prescribing the Proper Type and Doses of Vitamin D

Since vitamin D is a nutrient, the benefits from supplements are evident only in those with absolute or relative deficiencies. Therefore, while broader guidance on vitamin D supplementation is generally helpful, individual recommendations must be based on their vitamin D status, body weight, and requirements [10,20]. Therefore, as with other medical disorders, healthcare workers should investigate patients’ history, background, habits, and needs before recommending or prescribing vitamin D supplements. Along with lifestyle changes, this would allow them to advise individuals to take the correct dose, eliminate rare adverse effects, and reduce costs.

Access to a laboratory and affordable testing allows one to measure biochemical variables, such as serum 25(OH)D levels (and serum and urinary calcium, PTH, etc.), to assess the overall vitamin D/calcium status. Properly utilizing these data could provide cost-beneficial approaches for patients and maximize efficacy while minimizing adverse effects [160]. Considering the high cost of 25(OH)D measurements and their unavailability to most people worldwide, laboratory testing is not essential [20]. Instead of measuring serum 25(OH)D concentration, the proper amount of vitamin D is illustrated using body-weight-(or BMI)-based calculations using published tables in Nutrients [Wimalawansa, SJ, Nutrients, 14(14), 2997, 2022; https://doi.org/10.3390/nu14142997] [10,20].

Vitamin D is highly economical and has virtually no adverse effects on recommended D_3_ doses, frequency, and durations. Furthermore, it is widely available globally as an over-the-counter nutrient at an affordable cost. Maintaining vitamin D sufficiency in the population is the most cost-effective approach to controlling the spread of viral infections such as COVID-19 [20,98] and reducing the prevalence of chronic diseases [14,48,73,161] and healthcare costs.

### 4.6. Vitamin D Doses to Maintain Therapeutic Serum 25(OH)D Concentrations

For a healthy 70 kg non-obese adult, it is recommended to consume 5000 IU (125 micrograms)/day or 50,000 IU (1.25 mg: one capsule) every tenth day or once a week—especially during high-risk periods [20]. Those who are obese or have significant malabsorption require several fold-higher doses than the above. These doses could take several months for a vitamin D-deficient person to increase serum 25(OH)D to therapeutic levels of over 50 ng/mL [20]. Even in a vitamin D-sufficient person [i.e., based on community guidelines of maintaining serum 25(OH)D concentration of 40 ng/mL], the mentioned oral doses of vitamin D would take a few weeks to raise serum 25(OH)D concentration above 50 ng/mL [10]. Therefore, such doses could be insufficient (and ineffective) to achieve the desired target serum 25(OH)D concentration in emergencies.

Even for those with severe vitamin D deficiency, delays in raising serum 25(OH)D concentrations can be reduced to less than four days by administering higher upfront loading doses—between 100,000 IU and 400,000 IU based on the body weight, as a single dose [162]. If the time is not critical, higher amounts should preferably be administered in divided doses over a few days to improve absorption and aid 25-hydroxylation in the liver—a rate-limiting factor [20,163]. In contrast, administration of 0.5 to 1 mg calcifediol orally (preferably 0.014 mg/kg body weight) [20] rapidly increases serum 25(OH)D concentrations. Administration of the correct dose will boost the immune system within one day. Therefore, the latter is the most appropriate type of vitamin D in emergencies such as COVID-19 [10].

### 4.7. Vitamin D Dose Recommendations

Little vitamin D is present in natural food; thus, the dietary intake is minimal and cannot be relied upon for the majority [164,165]. Furthermore, without sufficient exposure to daily direct sunlight, casual exposure to the sun is insufficient to raise serum 25(OH)D concentration [14,166]. Governments and some scientific committees such as IoM, Food and Nutrition Board (FNNB), and USPTO in the USA, and the scientific advisory committee (SCAN) in the UK, etc., continue to recommend doses of vitamin D of between 400 and 1000 IU/day. These low doses would raise serum 25(OH)D concentration in the circulation to less than 6 ng/mL from their baseline, which is grossly insufficient in those with vitamin D deficiency and insufficiency [167]. Therefore, such doses consistently fail to raise serum 25(OH)D concentrations to needed therapeutic levels [167], thus making them unhelpful.

Governments and appointed committees recommend doses of vitamin D to prevent rickets in children and osteomalacia in adults [14,73,161], not for other disorders. It is only necessary to raise serum 25(OH)D concentration to 20 ng/mL to benefit the musculoskeletal system [16,168]. However, this does not help other body systems or disease conditions. Therefore, it is paramount to use adequate doses of vitamin D (preferably body-weight-based doses) [10,20] to increase serum 25(OH)D to the desired concentrations, as shown in Figure 4.

For busy healthcare workers, it is difficult to remember the different doses of vitamin D and serum 25(OH)D concentrations needed for various diseases [129,169]. Therefore, irrespective of age and body weight, when laboratory measurements are affordable and available, it is rational to maintain serum 25(OH)D concentrations above 40 ng/mL [170,171], preferably above 50 ng/mL—a range between 50 and 90 ng/mL [10,20,167].

When 25(OH)D measurements are unavailable or unnecessary, an accepted vitamin D dose for extra-skeletal beneficial effects of “non-obese” adults is 5000 IU per day (125 µg/day) [172,173]. In the vast majority, this would raise serum 25(OH)D concentration to approximately 40 ng/mL over time. These doses exceed outdated governmental recommendations [143,173,174]. Persons with obesity/overweight and multiple comorbidities, taking medications that increase 24-hydroxylase catabolic activity, and those with gastrointestinal fat malabsorption disorders require two- to four-fold higher doses than the above [10].

### 4.8. Strength and Limitations

Vitamin D is a vast field of science that encompasses all body systems. Over 120,000 peer-reviewed scientific articles on vitamin D have been published on PubMed alone, covering several aspects of its biology, physiology, clinical characteristics, and pathology related to vitamin D deficiency. Therefore, neither a single article nor a book could cover all aspects. This article focuses on physiological, clinical, and practical elements to help practicing clinicians and healthcare workers better manage their patients. These concepts were strengthened by providing pertinent references under each sub-heading for further reading and exploring other areas of interest.

## 5. Conclusions

Vitamin D_3_ supplements are widely available and inexpensive. It is a safe micronutrient that should be considered as a long-term prophylactic agent or adjunct therapy. When available and practical, direct and safe skin exposure to sunlight is preferable to obtain vitamin D_3_. Vitamin D deficiency is associated with many common chronic diseases, such as rickets, osteomalacia, metabolic and immune disorders, autoimmune diseases, and cancer. It also impacts pregnancy and children’s growth, including brain functions: the lack of it could lead to premature death. Consequently, its deficiency significantly increases the vulnerability and severity of metabolic diseases such as diabetes, obesity, and metabolic syndrome and worsens infections and cancer.

In addition, hypovitaminosis D increases the risks for bacterial and viral infections, associated complications, and deaths. Those affected are particularly vulnerable to viral respiratory illnesses such as seasonal flu, influenza, respiratory syncytial virus, and COVID-19. Worldwide, sepsis is responsible for over seven million annual deaths—a major contributory factor for this is vitamin D deficiency [175]. Half of these premature deaths can be prevented by healthcare workers taking proactive actions to treat patients aggressively, with vitamin D and calcifediol at the first encounter, as described in this article. The lack of utilization of scientific guidelines and recommendations for this highly cost-effective measure to benefit the public is striking. The goal is to rapidly raise serum 25(OH)D concentration above 50 ng/mL, allowing it to boost the immune system naturally [20]. This would have cost a fraction of the cost of managing these patients’ one-day hospital stay (less than two USDs per person).

This principle applies to those infected with or PCR-positive persons for SARS-CoV-2 (during the COVID-19 pandemic and in the future) [64,124,176]. Since February 2020, over 900 peer-reviewed scientific articles related to vitamin D and COVID-19 have been published—the first recommendation of using high doses of vitamin D was published on 28th February 2020 [177,178]. Based on the vast literature mentioned, vitamin D’s importance in rectifying the impairment of immune functions to overcome COVID-19 has recently been highlighted [179,180].

The long-term use of adequate vitamin D doses, such as 5000 to 7000 IU (125 to 175 micrograms)/day, will reduce the risks for infections and complications [57,173,181], racial disparities in infections-related clinical outcomes, including COVID-19 [124], and the spread of COVID-19, hospitalization and mortality [64]. Children and elders with severe hypovitaminosis D have a high risk of developing life-threatening infections [182]—conditions, such as Kawasaki-like disease or multi-system inflammatory syndrome [40,41,42], which can be significantly reduced by proactively providing vitamin D supplements.

In seriously ill patients (e.g., non-traumatic patients admitted to ICUs) with vitamin D deficiency, even high doses would take up to a week to increase serum 25(OH)D concentrations. For such patients and in emergencies, it is best to treat them with a single oral dose of 0.5 and 1.0 mg calcifediol (0.014 mg/kg body weight) to rapidly raise serum 25(OH)D concentrations above the minimum therapeutic levels of 50 ng/mL [20]. This dose can continue every third day until a high amount of vitamin D maintains the serum 25(OHD concentration at the desired level. Within four hours of administration of a proper dose of oral calcifediol, serum 25(OH)D concentrations increase above 50 ng/mL, sufficient to boost the immune system within one day.

The administration of high doses of vitamin D and/or calcifediol causes robust immune responses, allowing individuals to overcome severe inflammation and oxidative stress, thus preventing cytokine storms and ARDS [183]. Magnesium deficiency increases the risk and worsens cytokine storm-related complications and outcomes [184]. Therefore, it is crucial to prevent (intracellular) magnesium deficiency and provide sufficient amounts of other micronutrients and co-factors, allowing robust immune and metabolic functions. Because of the multiple biological and physiological benefits, the prompt administration of proper vitamin D dose supplements is necessary to prevent complications, ICU admissions, and death from infections [185]. Serum 25(OH)D concentration over 50 ng/mL is also necessary to overcome some cancers and reduce all-cause mortality.

Vitamin D supplementation prevents the development of chronic diseases [186,187], including osteoporosis and metabolic conditions [188] and acute and chronic respiratory infections [97,98,152]. SARS-CoV-2 infection, post-vaccinations, and other coronaviral infections are classic examples of where proper doses of vitamin D leading to sufficiency would prevent pulmonary and endothelial cell damage, hypoxia and ARDS, coagulation abnormalities and micro-embolism [82]. Consequently, it reduces hospitalizations, ICU admissions, and deaths [64,152,189]. In addition, vitamin D supplements at proper doses reduce incidences of seroconversion and symptomatic SARS-CoV-2 [125] and minimize complications [182], hospitalizations [190], ICU admissions [191], and deaths [64,189].

As illustrated in Section 4.7, the doses recommended (e.g., vitamin D, 400 to 1000 IU/day) by governments and some appointed advisory bodies were designed only to overcome the minimum serum 25(OH)D concentrations necessary to overcome musculoskeletal disorders like rickets in children and osteomalacia adults. Such low doses do not benefit other body systems. Despite these, none of these recommendations specify that the suggested minuscule amounts are only applied to prevent musculoskeletal diseases—rickets and osteomalacia.

Because of this narrow focus, the suggested doses and serum 25(OH)D concentrations are misleading and outdated. Considering the above, recommendations and broader conclusions by the entities mentioned above are misguiding [168,192] and cannot be generalized. Outdated recommendations and vitamin D guidelines must be urgently replaced with new documents using recently published data with solid and valuable guidance. Based on the vast amount of published data on vitamin D, those who write systematic reviews and meta-analyses must stop mindlessly repeating the jargon that “data are insufficient, and more RCTs are needed”.

Vitamin D metabolites or synthetic analogs of vitamin D should not be used as prevention or replacement therapy for deficiency and supplements or treatment for osteoporosis and infections. Vitamin D analogs are restricted to specific indications, such as hepatic or renal failure, hypoparathyroidism, etc. Calcifediol is the agent of choice in emergencies, where the rapid elevation of serum 25(OH)D concentration is necessary to reduce complications that save lives. Calculating calcifediol and vitamin D doses on a body-weight basis is straightforward [20]—thus, an individual’s required vitamin D dose should not be guessed. Calcifediol is administered as a single dose (in conjunction with a high amount of vitamin D to prolong the benefits of vitamin D) or every third day during an emergency. In addition, calcifediol is indicated when 25-hydroxylation of vitamin D is impaired, as in hepatic failure. However, it is not recommended as a replacement therapy for osteoporosis and renal failure. Instead, D_3_ should be used to correct vitamin D deficiency.

Furthermore, vitamin D metabolites and 1α-hydroxylated synthetic vitamin D analogs, including 1,25(OH)D, are indicated when renal tubular cells fail to generate calcitriol from 25(OH)D due to a lack of 1α-hydroxylation activity, as in advance renal failure. Calcitriol is also indicated in all types of hypoparathyroidism to maintain serum-ionized calcium. Treating deficiencies with nutrients to overcome conditions requires understanding biology and physiology in conjunction with practicalities and cost-effectiveness. With such an understanding, proper dosing regimens and the right frequency of administering nutrients using the three commercially available vitamin D metabolites (i.e., parental D_3_, calcifediol, and calcitriol) are crucial for improving clinical outcomes, avoiding adverse effects, and saving lives.

## Figures and Tables

**Figure 1 biomedicines-11-01542-f001:**
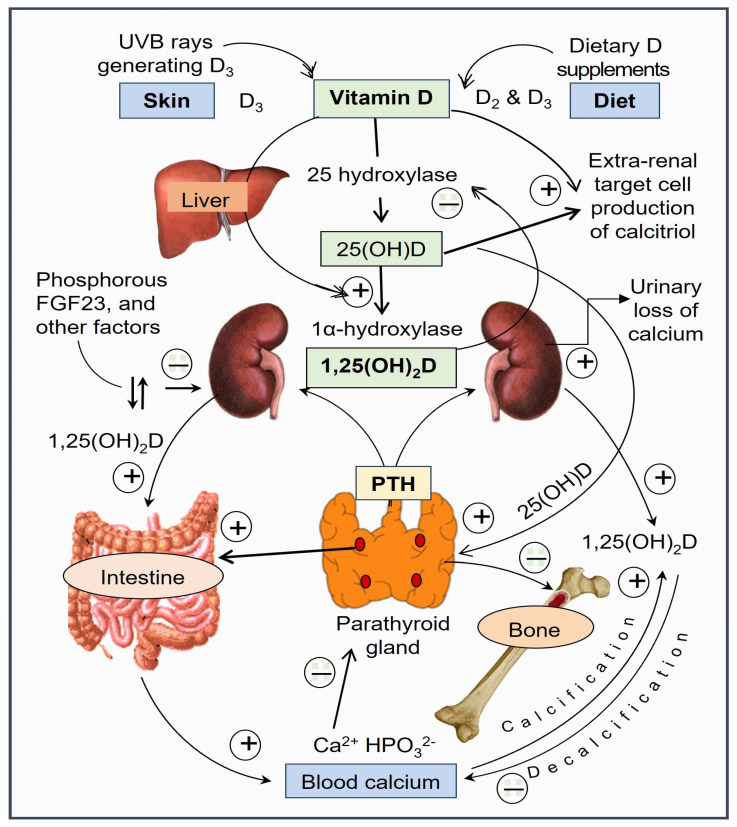
Schematic illustration of vitamin D generation and calcitriol’s calcium/mineral regulatory functions in conjunction with parathyroid hormone (PTH) and fibroblast growth factor-23 (FGF-23). The figure also depicts sites of activation of vitamin D—25-hydroxylation in the liver and 1α-hydroxylation in renal tubular cells and peripheral target cells. 1α-hydroxylation of 25(OH)D in proximal renal tubular cells generates the circulatory, hormonal form of calcitriol that controls calcium homeostasis.

**Figure 2 biomedicines-11-01542-f002:**
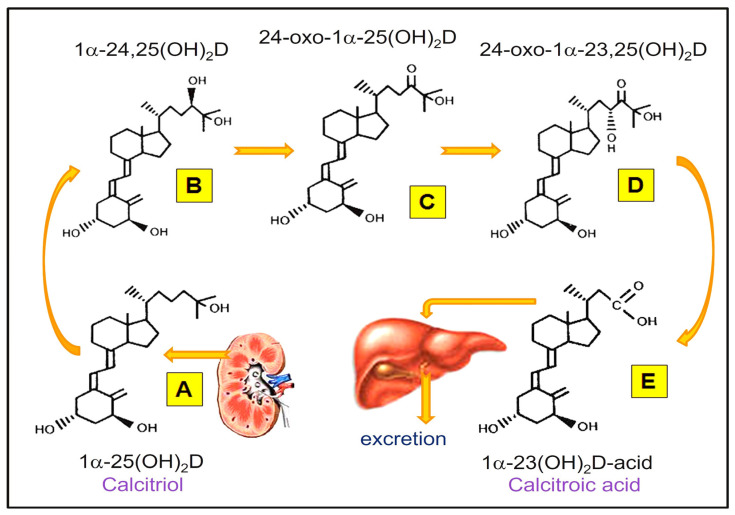
Common catabolic products of 1,25(OH)D_2_ (calcitriol). The inactive metabolic products of calcitriol and its excretory products, calcitroic acid, are illustrated.

**Figure 3 biomedicines-11-01542-f003:**
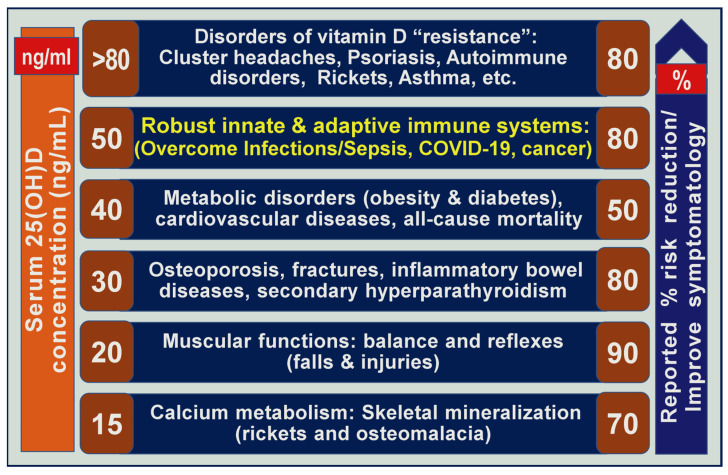
Illustrates calculated serum 25(OH)D concentrations needed to overcome different groups of conditions and disorders and the reported average (percentage) improvements/responses in primary clinical outcome. The figure summarizes cumulated data from many outcome-based vitamin D-related clinical studies.

**Figure 4 biomedicines-11-01542-f004:**
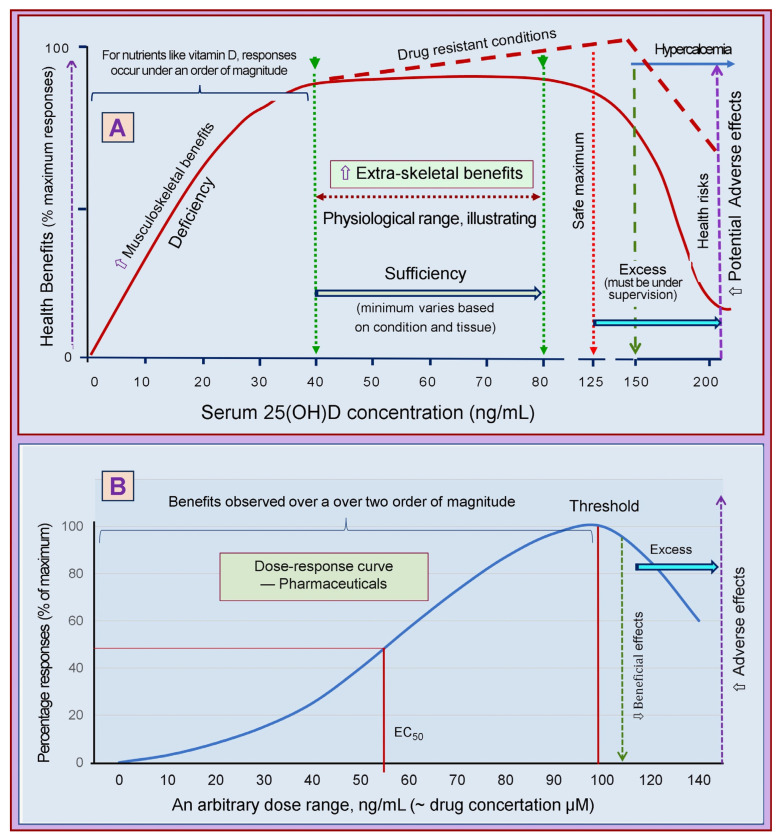
Illustrates pharmacodynamic differences and dose-response curves between nutrients and pharmaceutical agents. While nutrients show an abrupt response, dose-response curves of pharmaceutical agents spread over a broader range. (**A**) Depicts an example of vitamin D and dose responses. Providing more would not have additional physiological benefits when it reaches sufficiency for a given tissue/system. Furthermore, the response range is narrow, about half an order of magnitude. (**B**) The response range expands with pharmaceutical agents over an order of magnitude, and the response curve is shallow. The broken red line illustrates that the beneficial effects of vitamin D could continue without causing hypercalcemia when high doses are administered with very low calcium intakes and under close medical supervision.

**Table 1 biomedicines-11-01542-t001:** Key nutrient clinical study recruitment criteria: Clinical studies should be conducted to test the hypothesis that intervention (e.g., vitamin D supplements) benefit recipient subjects.

The Goal	Action Needed
Recruit only vitamin D-deficient subjects	Measurement of baseline serum 25(OH)D concentration. Recruit subjects with 25(OH)D serum levels less than 20 ng/mL (50 nmol/L) for clinical studies and RCTs.
Sufficient sample size	Based on statistical power calculation (on the effect size and the standard error of the mean).
Sufficient doses and the right frequency of administration (e.g., daily or weekly—not monthly or semi-annually)	Avoid administration of vitamin D at a frequency or less than once in two weeks. Age-appropriate proper doses must be used. Use the appropriate vitamin D dose to raise serum 25(OH)D concentration to a sufficient (target) level to achieve the intended outcome.
Ensure the sufficiency of co-nutrients and co-factors	For optimal function, supplemented nutrients interact and act synergistically with other nutrients. In the case of vitamin D (or calcium), ensure the availability of co-factors and supporting elements, such as magnesium, vitamin K_2_, etc.
Ensure the desired blood concentration is achieved (e.g., 25(OH)D concentration)	In longer trials, serum 25(OH)D concentrations should be measured after initiation of the intervention (e.g., approximately in four months).
Sufficient duration of the study	Shorter trials (e.g., acute infections) that last a few weeks vs. longer trials. Whereas chronic diseases, such as metabolic disorders, cancer, and osteoporosis, require several years of follow-up.
Keep the study clean	Study subjects should not take additional doses of index nutrients, including multivitamins, which might provide more of the same nutrients. Importantly, nutrients like vitamin D should not piggyback on pharmaceutical trials.
Keep the study simple	Use a simpler (uncomplicated) protocol with fewer variables and minimum number of study groups necessary to test the hypothesis. This decreases the number of subjects needed, improves statistical power, and makes for more straightforward interpretations and meaningful conclusions.
Clinical and statistical meaningfulness	Clinical study protocol must test a hypothesis based on a clinically meaningful increase in the indexed nutrient in circulation [i.e., 25(OH)D]—achieving and maintaining the blood levels above the minimum target is essential.
Balanced randomizations	Minimize confounders.
Target serum 25(OH)D concentration (and the therapeutic window)	Supplementation should bring serum 25(OH)D levels to a sufficiency level (at least above 40 ng/mL; 100 nmol/L) (in the case of infections, cancer, and autoimmune diseases, above 50 ng/mL)—the target goal of the clinical study.
Maintain a sustained effect	Maintain circulatory 25(OH)D concentrations above the target level during the entire study period in the interventional group [106].
Have firm-hard endpoints	The protocol should define hard primary endpoint(s)—e.g., complications such as reduced fractures, number needed to treat (NNT) to save one life, hospitalizations, ICU admissions, or deaths.
Over-the-counter nutrients, especially the index nutrient (e.g., vitamin D), and supplements, and vitamins should not be allowed to be taken during clinical studies [105,107,108]	Allowing patients to consume micro-nutrients from such a source will reduce the difference between the groups (the effect size) and the statistical power.
Statistical analyses	Correlation should be made with serum 25(OH)D concentrations achieved (active vs. control group) after supplements consumption [or at least, the changes (∆) from the baseline], but not with the administered dose.

## Data Availability

Not applicable.

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
