# Peer review of "Physiological Basis for Using Vitamin D to Improve Health"

_biomedicines, 2023, doi:10.3390/biomedicines11061542_

Round 1

Reviewer 1 Report

In this review, authors summarized the Physiological Basis for Using Vitamin D to Improve Health in depth. This review is valuable and will be of interest for a broad readership. It is recommended to be published in this journal after minor revision. The following questions need to be addressed.

1. The rationale of writing this review should be described in the introduction portion for the ease of the reader as well.

2. There are some grammatical and spelling errors in the manuscript that should be checked. For instance; In Section 1.1 line 63; concertation should be concentration and In Section 1.2 line 80; “Synthetic analogs of vitamin D are more than twenty-fold more expensive than vitamin D, so as the potential adverse effects” is unclear. The whole line or paragraph should be reconsidered as something is missing here or sentence seems to be incomplete. Moreover, the abbreviation of VDR that is Vitamin D receptors is missing from the whole manuscript.

3. A table should be included in the manuscript summarizing all clinical trials related with Vitamin D with its alternative forms and loop holes in the same for making manuscript more appetizing and for the ease of the reader.

The manuscript has some grammatical errors. 

Author Response

Reviewer 1:

  1. The rationale of writing this review should be described in the introduction portion for the ease of the reader as well.
     The author has added statements in this regard

Response:

  1. There are some grammatical and spelling errors in the manuscript that should be checked. For instance; In Section 1.1 line 63; concertation should be concentration and In Section 1.2 line 80; “Synthetic analogs of vitamin D are more than twenty-fold more expensive than vitamin D, so as the potential adverse effects” is unclear.

Thank you—The author has attended to these issues now.

The whole line or paragraph should be reconsidered as something is missing here or sentence seems to be incomplete. Moreover, the abbreviation of VDR that is Vitamin D receptors is missing from the whole manuscript.

Thanks - these have now been attended to—I appreciate constructive comments.

  1. A table should be included in the manuscript summarizing all clinical trials related with Vitamin D with its alternative forms and loop holes in the same for making manuscript more appetizing and for the ease of the reader.

Thank you for the suggestion: As rightly suggested, the author considered creating a Table but recognized that it would duplicate many of the references and descriptions already provided in the manuscript.

The author added a detailed table illustrating how to avoid serious errors in designing nutrition clinical trials, which will likely help the readers.

Thank you very much for your great input.

Reviewer 2 Report

The manuscript was prepared very well. The introduction section justifies the purpose of the study. I congratulate the authors for the preparation of the manuscript

I would like to congratulate the authors for the structure of the manuscript and all the research carried out. It is highly publishable. However, there are some concerns, in part important, so the review articles need revision, see below.

Introduction

-          Why is this study considered relevant?

-          Why is this study necessary?

Methods

-          Please, include the methodology of how you have related the search of the records used.

Discussion

·         Include a section on strengths / limitations.

·         There is a lack of a comparative discussion with other studies and an opinion/discussion of the authors

·         What mechanisms of action support these findings?

·         What does this article contribute to, the authors should make their own assessment and include their own discussion of the results shown in the manuscript?

·         Include some of the genetic variants associated with vitamin D https://doi.org/10.3390/ijms231911846

Conclusion

In the Conclusion section, state the most important outcome of your work. Do not simply summarize the points already made in the body — instead, interpret your findings at a higher level of abstraction. Show whether, or to what extent, you have succeeded in addressing the need stated in the Introduction (or objectives).

NA

Author Response

(Author's responses are in bold)

Improve the introduction and research design.

Thanks—the author has put in the effort it improves the introduction and the research design sections.

Improve conclusion to support results.

The conclusion section was modified to address the above – thanks again.

The manuscript was prepared very well. The introduction section justifies the purpose of the study. I congratulate the authors for the preparation of the manuscript.

Thank you very much for your great input.

I would like to congratulate the authors for the structure of the manuscript and all the research carried out. It is highly publishable. However, there are some concerns, in part important, so the review articles need revision, see below. 

Thank you

Introduction

-          Why is this study considered relevant?    Why is this study necessary?

     The author added a new sub-section (#1.5) to explain those mentioned above two important items.

Methods

-          Please, include the methodology of how you have related the search of the records used. 

     While important for a methodology paper, this suggestion may not be relevant to this manuscript.

Discussion

  • Include a section on strengths / limitations.

     The author added section, 4.7 to discuss the strength and limitations of this manuscript – Thank you for the suggestion.

  • There is a lack of a comparative discussion with other studies and an opinion/discussion of the authors. What mechanisms of action support these findings?
  • What does this article contribute to, the authors should make their own assessment and include their own discussion of the results shown in the manuscript.

     The author has attempted to address this with the limit of not exceeding word counts.

  • Include some of the genetic variants associated with vitamin D https://doi.org/10.3390/ijms231911846

     The author has added a new sub-section (#4.4) to briefly discuss gene polymorphisms and the epigenetic influences of vitamin D metabolites and VDR interactions.

Conclusion

In the Conclusion section, state the most important outcome of your work. Do not simply summarize the points already made in the body — instead, interpret your findings at a higher level of abstraction. Show whether, or to what extent, you have succeeded in addressing the need stated in the introduction (or objectives).

Great suggestion. The author tried to achieve the mentioned goal, but it may not be complete due to certain constraints.

Thank you very much for your great input.

Reviewer 3 Report

This is a comprehensive review about the functions of vitamin D and the difficulty of identifying how good vitamin D status can protect against a wide range of diseases, when clinical trials do not have subjects that are uniformly of low vitamin D status. There are a number of points in the manuscript which are either unclear or need further information.

Lines 62-63: References are needed to support the stated concentrations of serum 25(OH)D found in people with plenty of exposure to the sun.

Line 126: “Giving vitamin D to those who are sufficiently unlikely to provide an additional benefit: is one of the fundamental errors…..” This sentence does not make sense as it is written.

Line 128: What is meant by the term “Right doses of vitamin D” ?

Line 130: What is meant by the term “high-density of vitamin D” ?

Line 160: What is meant by the term “longer lasting” ?

Line 162: “increases the expression of 1α-hydroxylase” – No, the enzyme is a 25-hydroxylase.

Line 164: The enzyme which is the 25(OH)D-1-hydroxylase is CYP27B1 not CYP2R1

Line 180: The enzyme is CYP27B1, not CYP2R1

Line 322: It would help to explain the differences between the blood concentrations of pharmaceutical agents and those of metabolites of vitamin D if it was pointed out that storage mechanisms have evolved for vitamin D and 25(OH)D which explain the long half-life of 25(OH)D in blood. No storage mechanisms would apply to pharmaceutical agents.

Lines 358-362: References are needed to justify the claims for the length of time of oral intake of vitamin D and the response in 25(OH)D serum concentrations. An oral dose of 5000 iu per day (122 µg/day) is considerably greater than most people would obtain from vitamin D found naturally in food.

Explained in comments to the author

Author Response

(Author's responses are in bold)

Referee 3:

This is a comprehensive review about the functions of vitamin D and the difficulty of identifying how good vitamin D status can protect against a wide range of diseases, when clinical trials do not have subjects that are uniformly of low vitamin D status. There are a number of points in the manuscript which are either unclear or need further information.

Lines 62-63: References are needed to support the stated concentrations of serum 25(OH)D found in people with plenty of exposure to the sun.

Done – Thank you

Line 126: “Giving vitamin D to those who are sufficiently unlikely to provide an additional benefit: is one of the fundamental errors…..” This sentence does not make sense as it is written.

These two sentences were modified to make the message clear.

Line 128: What is meant by the term “Right doses of vitamin D” ?

This sentence was rewritten to make it clear.

Line 130: What is meant by the term “high-density of vitamin D” ?

This sentence was also rewritten to make it clear.

Line 160: What is meant by the term “longer lasting” ?

Thank you—this sentence was also rewritten to make it clear.

Line 162: “increases the expression of 1α-hydroxylase” – No, the enzyme is a 25-hydroxylase.

Line 164: The enzyme which is the 25(OH)D-1-hydroxylase is CYP27B1 not CYP2R1

Thank you for identifying the above two mix up—it was corrected.

Line 180: The enzyme is CYP27B1, not CYP2R1

Done – thank you.

Line 322: It would help to explain the differences between the blood concentrations of pharmaceutical agents and those of metabolites of vitamin D if it was pointed out that storage mechanisms have evolved for vitamin D and 25(OH)D which explain the long half-life of 25(OH)D in blood. No storage mechanisms would apply to pharmaceutical agents.

Great point—the author has expanded the paragraph illustrating the above.

Lines 358-362: References are needed to justify the claims for the length of time of oral intake of vitamin D and the response in 25(OH)D serum concentrations. An oral dose of 5000 iu per day (122 µg/day) is considerably greater than most people would obtain from vitamin D found naturally in food.

Thank you for this important recommendation—the author has now attended to it by adding an entire paragraph in this regard.

Thank you very much for your great input.

Round 2

Reviewer 2 Report

The authors have made substantial changes to the manuscript and have improved its quality. I congratulate you. However, the language needs to be improved. In addition, it should include at least 1 paragraph of the author's own creation where he includes that he has contributed to this manuscript. In addition, the conclusion part should be improved, highlighting the most relevant findings.

the language should be checked by a native person

Author Response

Biomedicine/MDPI

Referee 2:

The authors have made substantial changes to the manuscript and have improved its quality. I congratulate you. However, the language needs to be improved. In addition, it should include at least 1 paragraph of the author's own creation where he includes that he has contributed to this manuscript. In addition, the conclusion part should be improved, highlighting the most relevant findings.

Revisions and editing were done to e improve English and presentation.

Referee:  As one paragraph of the author's own creation where he includes that he has contributed to this manuscript.

Response:  Thank you: Much appreciated the input.

Line 1.6, Section 1.6
The author added a paragraph at the end of section #1.5 (just before the beginning of section #2.0 to address the above.

Referee:  In addition, the conclusion part should be improved, highlighting the most relevant findings.

Line 632, Section 5.0
Response: Thank you so much: The conclusion section has been modified and improved.

The author also improves the flow and the content in the Introduction and Section #1